# Clinical Instability at Discharge and Post-Discharge Outcomes in Patients with Community-Acquired Pneumonia: An Observational Study

**DOI:** 10.3390/jcm14155273

**Published:** 2025-07-25

**Authors:** Yogesh Sharma, Arduino A. Mangoni, Rashmi Shahi, Chris Horwood, Campbell Thompson

**Affiliations:** 1Department of Acute and General Medicine, Flinders Medical Centre, Adelaide 5042, Australia; 2College of Medicine & Public Health, Flinders University, Adelaide 5042, Australia; arduino.mangoni@flinders.edu.au (A.A.M.); rashmishahi5@hotmail.com (R.S.); 3Clinical Improvement Unit, Flinders Medical Centre, Adelaide 5042, Australia; chris.horwood@sa.gov.au; 4Discipline of Medicine, University of Adelaide, Adelaide 5005, Australia; campbell.thompson@adelaide.edu.au

**Keywords:** community-acquired pneumonia, clinical instability, mortality, readmissions

## Abstract

**Background/Objectives**: Clinical stability within 24 h prior to discharge is a key metric for safe care transitions in hospitalised patients with community-acquired pneumonia (CAP). However, its association with post-discharge outcomes, particularly readmissions, remains underexplored. This study assessed whether clinical instability before discharge is associated with 30-day mortality, readmissions, or a composite of both in hospitalised CAP patients. **Methods**: This retrospective cohort study included adults (≥18 years) admitted with CAP to two tertiary Australian hospitals between 1 January 2020 and 31 December 2023. Clinical instability was defined as abnormal vital signs (temperature, heart rate, respiratory rate, blood pressure, or oxygen saturation) within 24 h before discharge. Pneumonia severity was assessed using the CURB-65 score and frailty using the Hospital Frailty Risk Score. Multilevel logistic regression models were used to evaluate associations with outcomes, adjusting for age, sex, comorbidities, frailty, disease severity, microbiological aetiology, antibiotics prescribed during admission, and prior healthcare use. Competing risk regression accounted for death when analysing readmissions. **Results**: Of 3984 patients, 20.4% had clinical instability within 24 h before discharge. The composite outcome occurred in 21.9% patients, with 15.8% readmitted and 6.1% dying within 30 days. Clinical instability was significantly associated with the composite outcome (adjusted odds ratio [aOR] 1.73, 95% CI 1.42–2.09, *p* < 0.001), primarily driven by increased mortality risk (aOR 3.70, 95% CI 2.73–5.00, *p* < 0.001). However, no significant association was found between clinical instability and readmissions (aOR 1.16, 95% CI 0.93–1.44, *p* > 0.05). **Conclusions**: Clinical instability within 24 h before discharge predicts worse outcomes in CAP patients, driven by increased mortality risk rather than readmissions.

## 1. Introduction

Community-acquired pneumonia (CAP) remains one of the most common and serious causes of hospitalisation globally, frequently leading to significant post-discharge morbidity and mortality [1]. A recent Australian study reported a 30-day mortality rate of 17% and a 30-day readmission rate of 21% [2]. Over recent decades, reductions in hospital length of stay (LOS) for CAP and other acute medical conditions have raised critical concerns about the safety of discharge practices. Specifically, premature discharge in clinically unstable patients is increasingly recognised as a major contributor to adverse outcomes, including avoidable readmissions and mortality [3,4].

Approximately 20% of avoidable readmissions among general medical patients and 10% among those hospitalised for CAP are attributed to premature discharge [5,6]. These findings highlight an urgent need to assess current protocols and ensure that clinical stability is achieved before hospital discharge. Current clinical guidelines recommend that resolution of clinical instability should guide discharge decisions, as this can serve as a simple, objective measure of safety [7,8]. For instance, the 2024 Dutch Working Party on Antibiotic Policy (SWAB)/NVALT CAP guideline [9], the NICE guideline on pneumonia [10], and clinical frameworks such as those from Intermountain Health [11] and Medscape [12] all emphasise ensuring physiological stability and appropriate clinical recovery before discharge. These guidelines increasingly recognise the importance of not only antimicrobial optimisation but also a holistic assessment of discharge fitness based on vital signs, functional status, and comorbidities.

Assessment of vital signs in the 24 h prior to discharge offers a practical and evidence-based approach to evaluate clinical readiness. Evidence from a large U.S. cohort study of over 32,000 hospitalisations underscores the importance of this approach [13]. The presence of one or more unstable vital signs at discharge was associated with significantly higher odds of death or readmission (aOR 1.36, 95% CI 1.26–1.48), with an even stronger association with mortality (aOR 2.31, 95% CI 1.91–2.79). Furthermore, the risk of adverse outcomes increased progressively with the number of unstable vital signs present. However, the study did not account for frailty—an established predictor of poor outcomes—and disease severity [14].

This issue has become even more pressing in the context of the COVID-19 pandemic, which has fundamentally altered patterns of hospitalisation, discharge practices, and post-discharge outcomes [15,16]. Disruptions in health services and evolving patient behaviours during the pandemic may have further exacerbated the risks associated with discharge instability. Additionally, most previous studies have limited their focus to readmissions within the index hospital, neglecting the 10–15% of readmissions occurring at non-index hospitals [17,18].

Therefore, we aimed to address these critical gaps by evaluating the impact of clinical instability—defined as abnormal vital signs within 24 h prior to discharge—on post-discharge outcomes in hospitalised patients with CAP. Unlike previous studies [13,19], we accounted for both CAP severity and frailty, two key determinants of clinical outcomes. We hypothesised that clinical instability within 24 h prior to discharge, irrespective of frailty status, is associated with significantly worse outcomes, including an increased risk of mortality and readmission.

## 2. Materials and Methods

### 2.1. Study Design and Data Sources

This retrospective cohort study was conducted across two major metropolitan hospitals in South Australia (Royal Adelaide Hospital and Flinders Medical Centre) and included all adult patients admitted with CAP between 1 January 2020 and 31 December 2023. CAP was defined as the presence of symptoms such as fever, cough (with or without sputum production), or dyspnea in conjunction with radiological evidence of pulmonary infiltrate [7]. Cases were captured using the International Classification of Diseases 10th Revision Australian Modification (ICD-10-AM) codes for CAP. Exclusion criteria were patients transferred to other acute care settings, those discharged against medical advice, and those who died during hospitalisation due to the increased likelihood of unstable vital sign observations in these groups. Data on demographic characteristics, clinical variables, and medical history were retrieved from electronic medical records (EMR). Pneumonia severity was assessed using the CURB-65 score, which was retrospectively calculated based on clinical and laboratory data obtained from the EMR [20]. Frailty status was determined using the Hospital Frailty Risk Score (HFRS) [21]. Patients with an HFRS ≥ 5 were classified as frail [22]. Information on the microbiological aetiology was obtained from culture results and viral polymerase chain reaction (PCR) testing. Data on antibiotics prescribed for the treatment of CAP were extracted from the pharmacy records of the hospitals. Variables previously identified as risk factors for readmissions, such as number of hospital admissions in the previous year and the number of emergency department (ED) visits within the six months preceding the index hospitalisation, were also included in the analysis [23]. Ethical approval for this study was granted by the Central Adelaide Local Health Network Clinical Ethics Committee (approval number 18887, 16 January 2024). As this was a retrospective observational study, individual patient consent was not required and was waived by the ethics committee.

### 2.2. Definition of Clinical Instability

Clinical instability was defined as the presence of any abnormality in vital signs recorded within the 24 h preceding discharge, including temperature, heart rate, respiratory rate, systolic blood pressure, and oxygen saturation. Vital sign abnormalities were determined based on clinically accepted thresholds, with clinical instability defined as the presence of any of the following: temperature ≥ 37.8 °C, heart rate ≥ 100 beats per minute, respiratory rate > 24 breaths per minute, systolic blood pressure ≤ 90 mm Hg, and oxygen saturation < 90%. These cut-off values are based on clinical face validity and are supported by prior literature [13,19]. Patients who lacked documentation of any vital signs within 24 h prior to discharge were excluded from the analysis. This exclusion is detailed in the study flowchart.

### 2.3. Outcomes

The primary outcome was a composite measure of death or readmission within 30 days of discharge. Patients who experienced both death and readmission within this period were counted only once for the composite outcome. Secondary outcomes included 30-day mortality and 30-day readmissions, considered separately. Readmissions were tracked for any public hospital within South Australia.

### 2.4. Statistical Analysis

Data were tested for normality using the Shapiro–Wilk test. Baseline characteristics were compared between patients with and without the composite outcome (death or readmission) using *t*-tests or ranksum tests for continuous variables, as appropriate, and chi-squared tests for categorical variables.

A random intercept multilevel logistic regression model was employed to assess the association between clinical instability and the composite outcome, with hospital specified as a clustering variable to account for potential differences in patient populations, care practices, and outcome rates across sites. The model was adjusted for potential confounders including age, sex, Charlson comorbidity index (CCI), CURB-65 score, HFRS, number of hospital admissions in the previous 1 year, number of ED presentations in last 6 months, microbiological aetiology, and antibiotics prescriptions. These covariates were selected based on prior literature, clinical relevance, and biological plausibility, given their established associations with adverse outcomes in patients with CAP [24,25,26].

To evaluate the appropriateness of the multilevel structure, we calculated the hospital-level variance component and the intraclass correlation coefficient (ICC). These measures quantified the proportion of total outcome variance attributable to between-hospital differences.

Odds ratios (OR) with 95% confidence intervals (CI) were computed. To assess the relationship between the number of abnormal vital signs and adverse outcomes, the Cochran–Armitage trend test [27] was used. An interaction term between the years of study and clinical instability was incorporated into the regression models to evaluate the confounding effect of the COVID-19 pandemic on observed clinical outcomes.

### 2.5. Sensitivity Analysis

We performed a competing risk regression analysis using the Fine and Gray model [28] to account for death as a competing risk for readmissions. Subdistribution hazard ratios (SHRs) with 95% confidence intervals (CIs) were calculated. The model was adjusted for the previously mentioned variables. Additionally, a cumulative incidence function (CIF) curve was plotted to illustrate the probability of readmission over time while accounting for the competing risk of death. All statistical analysis were performed using STATA software vs 19.0 (StataCorp LLC, College Station, TX, USA) and Python vs 3.12.4.

## 3. Results

Over a 4-year period, 6319 adults with CAP were admitted to the two hospitals (Figure 1). After applying exclusion criteria (including in-hospital deaths, transfer to another hospital, hospital at home, and self-discharge), 4677 patients remained. Of these, 693 patients (14.8%) had missing data on vital signs in the 24 h prior to discharge and were excluded. The final analytic sample comprised 3984 patients, of whom 813 (20.4%) experienced clinical instability within 24 h of discharge (Figure 1). Among those included, 374 patients (9.4%) had missing data for antibiotic prescriptions, and all other covariates were complete.

Among those with instability at discharge, 657 (80.8%) had one, 133 (16.4%) had two unstable observations, and 23 (2.8%) had three or more unstable observations. Tachycardia (10.7%) was the most frequent vital sign abnormality at discharge followed by temperature elevation (4.7%) (Appendix A). Eight hundred and seventy-six (21.9%) CAP patients experienced the composite outcome, defined as either readmission or death within 30 days of hospital discharge. Of these, 706 (17.7%) were readmitted, while 246 (6.2%) died within 30 days of discharge.

Patients who experienced a composite outcome were more likely to be older males with a higher number of comorbidities, greater frailty, and more severe CAP (*p* < 0.05). These patients also had a longer LOS during their index admission, as well as a higher number of hospitalisations in the previous year and increased ED presentations in the 6 months prior to their index admission (*p* < 0.05). However, microbiological aetiology and antibiotic prescription patterns were similar between the two groups (*p* > 0.05) (Table 1).

The composite outcome occurred more frequently among CAP patients with any clinical instability within 24 h prior discharge (248 (30.5%) vs. 628 (19.8%), *p* < 0.001). The likelihood of the composite outcome increased with the number of vital sign abnormalities; 73.9% of patients with three or more vital sign abnormalities experienced a composite outcome, compared to 28.0% of those with only one abnormality (*p* < 0.001) (Figure 2). While 30-day mortality also rose significantly with increasing number of vital sign abnormalities (*p* < 0.001), no significant difference was observed in readmission rates across groups (*p* > 0.05) (Figure 2).

While the median LOS decreased with the onset of the COVID-19 pandemic in 2020, it gradually increased in the subsequent years (Appendix A). The proportion of patients discharged with clinical instability mirrored this trend, showing no significant change over the years (*p* > 0.05). Similarly, no significant difference was observed in the composite outcome during this period (*p* > 0.05) (Appendix A).

Multilevel regression analysis revealed that patients with any clinical instability within 24 h prior to discharge had 1.73 times higher odds of experiencing the composite outcome (aOR 1.73, 95% CI 1.42–2.09, *p* < 0.001), after adjusting for age, sex (reference: male), CCI, frailty (reference: non-frail), CURB65 score, previous hospitalisations in the year prior, ED visits in the 6 months preceding admission, microbiological aetiology (reference: no pathogen detected) and antibiotics prescribed (reference: penicillin), while accounting for hospital level clustering (Table 2). The estimated hospital-level variance was negligible (variance = 3.35 × 10^−35^), with an ICC near zero. The likelihood ratio test comparing the multilevel model to a standard logistic regression showed no improvement in model fit (χ^2^ = 0.00), indicating that outcomes were not meaningfully clustered by hospital. 

While mortality was significantly higher among patients with clinical instability (aOR 3.70, 95% CI 2.73–5.00, *p* < 0.001) readmissions did not show a significant difference in the adjusted analysis compared to patients without clinical instability (aOR 1.16, 95% CI 0.93–1.44, *p* > 0.05), suggesting that clinical instability primarily influences mortality risk rather than readmission risk (Table 2).

The association of individual vital sign abnormalities and the composite clinical outcome is shown in Table 3. While hypotension (SBP ≤ 90 mm Hg) had the strongest association with the composite clinical outcome (aOR 4.51, 95% CI 2.51–8.11, *p* value < 0.001) followed by tachypnoea (RR > 24/min) (aOR 3.13, 95% CI 2.16–4.54, *p* < 0.001), temperature elevation was not associated with the composite clinical outcome (aOR 0.91, 95% CI 0.59–1.41, *p* = 0.699).

Sensitivity analysis showed a non-significant association trend between clinical instability within 24 h prior to discharge and readmission risk after adjusting for mortality (SHR 1.17, 95% CI 0.98–1.40, *p* = 0.067) (Figure 3).

## 4. Discussion

This multicentre cohort study of nearly 4000 patients hospitalised with CAP provides strong evidence that clinical instability in the 24 h prior to discharge is associated with significantly higher odds of composite outcome of 30-day death or readmission. Importantly, the risk of adverse outcomes increased progressively with the number of abnormal vital signs at discharge, reinforcing the value of clinical stability assessments in discharge planning. This study found that median LOS declined with the onset of the COVID-19 pandemic but increased in the subsequent years. No significant trends were observed in the proportion of patients discharged with clinical instability or the adverse composite outcomes during this period.

### 4.1. Key Findings and Comparison with Literature

The findings of this study aligns with previous research [13,29], which also reported that more than 20% of CAP patients are discharged with clinical instability and abnormal vital signs at discharge are associated with an increased risk of combined readmission and death within 30 days of discharge. However, in contrast to previous research [13], our study revealed no significant association between clinical instability and readmissions when the latter were analysed as an individual outcome. Previous studies [30,31] suggest that readmissions are a complex phenomenon which is difficult to predict and may be influenced by a range of factors such as functional and socio-economic status apart from medical issues. It is possible that readmissions following CAP hospitalisation are more likely influenced by comorbidities, frailty, and other underlying conditions rather than discharge instability [32]. If this hypothesis is true, then consideration of 30-day readmissions as a reliable indicator of quality of care for CAP patients is debatable. It is plausible that factors such as discharge communication, post-discharge follow-up, and home care interventions may play a more significant role in determining readmissions rather than clinical stabilisation [29,33]. Further studies are required to confirm or refute this proposition.

### 4.2. Vital Sign Abnormalities and Individual Outcomes

Similar to earlier studies [29,34], our findings suggest that the likelihood of adverse composite outcomes (readmission or death) increases with the number of unstable vital signs at discharge. Regarding individual clinical outcomes, our study identified hypotension and tachypnoea as having the strongest associations with poor outcomes, while elevated temperature showed no significant association. Hypotension in CAP patients may reflect severe underlying infection leading to sepsis or other factors such as dehydration, use of antihypertensive medications, or exacerbation of underlying cardiovascular disease, making it a critical predictor of poor outcomes [35,36,37]. Similarly, tachypnoea which could reflect severe pneumonia or its evolving complications such as empyema as well as non-infective causes, such as exacerbation of heart failure or development of pulmonary embolism [36,38]. Limited research has explored the relationship between individual vital sign abnormalities and clinical outcomes in CAP. Capelastegui et al. [39] found that elevated temperature within 24 h of discharge was strongly associated with short-term adverse outcomes including readmissions and death following CAP hospitalisation. This contrasts with our study, where temperature elevation had no significant relationship with clinical outcomes. One possible explanation is the difference in discharge practices between the two studies. In the study by Capelastegui et al., patients may have been discharged earlier, prior to achieving defervescence, as evidenced by a shorter median hospital stay (3 days versus 3.9 days in our study). This discrepancy underscores the need for further research into the role of individual vital sign abnormalities in predicting outcomes and highlights the importance of tailoring discharge readiness assessments based on multiple clinical parameters.

### 4.3. Temporal Trends and COVID-19 Context

This study suggests that hospital LOS initially declined with the onset of the COVID-19 pandemic, followed by a gradual increase in subsequent years. This trend occurred despite the significant rise in COVID-19-related hospitalisations. For instance, at the Royal Adelaide Hospital, the number of COVID-related admissions surged from 43 in 2021 to 3213 in 2022, and then declined slightly to 1242 in 2023 [25]. This major increase in COVID-19-related admissions reflects the evolving nature of the pandemic and the varying pressures placed on the healthcare system over time.

Interestingly, while previous research [34] has suggested that reduced LOS may contribute to increased discharge instability, our study did not find a significant relationship between shorter hospital stays and discharge instability during the early stages of the pandemic. This finding contrasts with the previous study that raised concerns about the potential risks of premature discharge, including poorer post-discharge outcomes. Our study findings are both reassuring and significant. They suggest that, despite the overwhelming pressure on hospital systems and the drive for efficiency during the pandemic, the reduction in LOS did not result in compromised discharge practices. It is likely that healthcare teams adapted their practices effectively to ensure safe and appropriate discharge planning, even under the strain of increased hospital admissions and constrained resources [40,41]. This finding reflects the resilience and adaptability of healthcare providers in maintaining the quality of patient care during challenging circumstances.

### 4.4. Implications for Discharge Planning and Post-Discharge Care

Our findings support current clinical guidelines recommending that discharge decisions be informed by vital sign stability. Recent international guidelines highlight objective resolution of vital sign abnormalities—especially respiratory rate and blood pressure—as essential for safe discharge decisions [9,10,11]. However, as readmissions may be less influenced by physiologic measures, multifactorial post-discharge strategies—including enhanced discharge planning, patient education, early follow-up, and community supports—are likely more effective at preventing readmissions [33,42]. These results reinforce the growing consensus that bundled transitional care interventions can reduce readmission risk more effectively than discharge criteria alone.

### 4.5. Strengths

Key strengths of this study include its large sample size, inclusion of multiple hospitals, and the use of comprehensive linked data that captured both in-hospital and post-discharge events across institutions. Unlike prior studies, we adjusted for frailty and disease severity and included a wide range of clinical variables, improving the validity of our findings.

### 4.6. Limitations

Our study has several limitations that should be considered when interpreting the findings. First, we defined clinical instability as the presence of any abnormal vital signs within the 24 h prior to discharge. While this provided a standardised approach, it may not fully capture the patient’s physiological status immediately prior to discharge, as some patients may have achieved clinical stability shortly, e.g., a few hours, before being discharged. Second, we did not exclude patients who were receiving home oxygen therapy, which could have affected baseline oxygen saturation levels, potentially influencing our assessment of clinical instability. Third, we were unable to identify patients who were discharged with palliative intent, particularly those transferred to nursing homes, who likely had a higher baseline risk of death. As this was a retrospective study, we were unable to control for all potential confounders, and there may have been factors that influenced patient outcomes that we were unable to account for, thus introducing the potential for unmeasured biases. Readmissions were only captured within the South Australian public hospital system. This may have led to under-ascertainment of readmission events among patients who were admitted to private hospitals or facilities outside the state, potentially underestimating the true rate of readmissions.

Additionally, our dataset lacked detailed information on certain established risk factors for CAP outcomes, such as smoking status, individual socioeconomic factors, and non-antibiotic medications (e.g., proton pump inhibitors, antipsychotics) that may influence pneumonia risk or recovery [42,43]. This may have limited our ability to fully adjust for all potential confounding variables.

## 5. Conclusions

This study suggests that clinical instability within 24 h prior to discharge in hospitalised CAP patients is a strong predictor of adverse composite outcomes, predominantly driven by mortality rather than readmissions. Strategies such as delaying discharge until clinical stability is achieved may help reduce short-term mortality in this population. However, the limited impact on readmission rates suggests the need to further investigate additional factors, such as frailty, to enhance post-discharge outcomes and optimise care transitions.

## Figures and Tables

**Figure 1 jcm-14-05273-f001:**
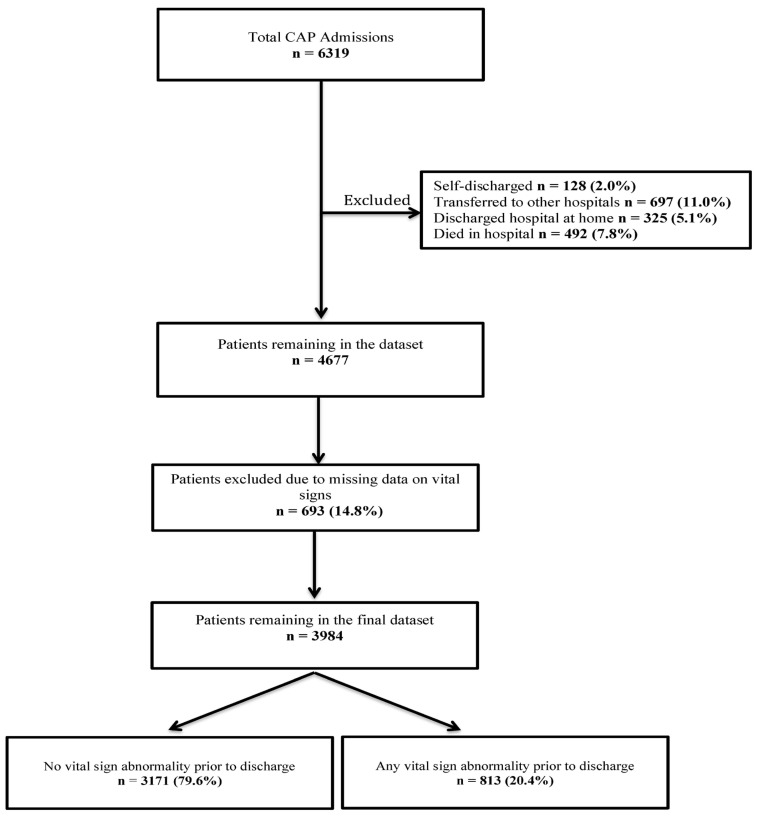
Study flow diagram.

**Figure 2 jcm-14-05273-f002:**
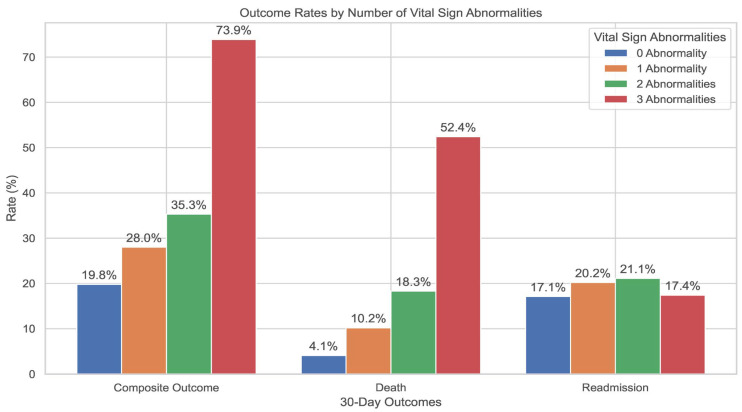
Rates of 30-day adverse outcomes stratified by number of vital sign abnormalities prior to discharge.

**Figure 3 jcm-14-05273-f003:**
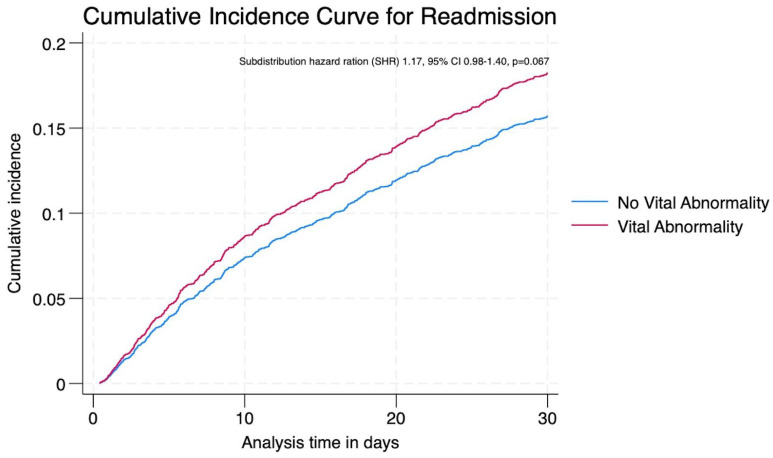
Cumulative incidence function (CIF) from competing risk regression showing 30-day readmission risk, accounting for death as a competing event.

**Table 1 jcm-14-05273-t001:** Characteristics of community-acquired pneumonia patients based on 30-day death or readmission.

Variable	Total Cohort	No Death or Readmission Within 30-Days	Death or Readmission
	3984	3108 (78.1)	876 (21.9)
Age years, mean (SD)	73.4 (17.7)	72.8 (18.1)	75.5 (16.1)
Age group, n (%)			
<40	254 (6.3)	216 (6.9)	38 (4.3)
40–59	521 (13.1)	428 (13.7)	93 (10.6)
60–79	1451 (36.4)	1137 (36.6)	314 (35.8)
>80	1758 (44.1)	1327 (42.7)	431 (49.2)
Sex male, n (%)	2186 (54.8)	1677 (53.9)	509 (58.1)
Residence home, n (%)	3768 (94.5)	2954 (95.1)	814 (92.9)
Charlson index, mean (SD)	2.6 (2.8)	2.2 (2.6)	3.8 (3.3)
CURB-65 score, mean (SD)	1.6 (1.1)	1.6 (1.1)	1.8 (0.9)
Severe CAP ^†^, n (%)	871 (21.9)	654 (21.0)	217 (24.8)
HFRS, mean (SD)	4.9 (4.5)	4.8 (4.4)	5.8 (4.7)
Frail, n (%)	1542 (38.7)	1140 (36.7)	402 (45.9)
Microbiological aetiology n (%)			
No pathogen detected	3382 (84.8)	2695 (84.9)	687 (84.5)
Bacterial	285 (7.2)	226 (7.2)	59 (7.3)
Viral	234 (5.9)	191 (6.0)	43 (5.2)
Polymicrobial	46 (1.2)	29 (0.9)	17 (2.1)
Others	37 (0.9)	30 (1.0)	7 (0.9)
Antibiotics prescribed n (%)			
Penicillin	1024 (25.7)	814 (25.6)	210 (25.8)
Cephalosporin	562 (14.1)	446 (14.1)	116 (14.3)
Macrolide	1465 (36.7)	1156 (36.5)	309 (38.1)
Quinolone	94 (2.4)	82 (2.6)	12 (1.5)
Doxycycline	285 (7.1)	234 (7.4)	51 (6.3)
Others	554 (13.1)	439 (13.8)	115 (14.2)
Hospital admissions past 12 months, mean (SD)	0.7 (1.4)	0.5 (1.3)	1.1 (1.8)
ED visits past 6 months, mean (SD)	0.5 (0.5)	0.5 (0.5)	0.7 (0.5)
LOS, median (IQR)	3.9 (2.3, 6.6)	3.8 (2.2, 6.3)	4.2 (2.5, 7.3)
30-day mortality, n (%)	246 (6.1)	0	246 (28.1)
^††^ 30-day readmissions n (%)	630 (15.8)	0	630 (71.9)

^†^ Severe CAP defined as CURB-65 score ≥3. ^††^ Readmissions excluded patients who died within 30 days to ensure events are counted only once within the composite measure. SD, standard deviation; CURB-65, (pneumonia severity score calculated from following parameters: confusion, urea levels >7 mmol/L, respiratory rate ≥30/min, blood pressure systolic <90 mm Hg or diastolic ≤60 mm Hg, and age ≥65 years); CAP, community-acquired pneumonia; HFRS, Hospital Frailty Risk Score; ED, emergency department; LOS, length of hospital stay; IQR, interquartile range.

**Table 2 jcm-14-05273-t002:** Relationship between vital sign abnormality before discharge and clinical outcomes after multilevel logistic regression model.

	Number of Unstable Vitals on Discharge					*p* Value
	Any	0	1	2	≥3	
Composite outcome						<0.001
Unadjusted Odds ratio (95% CI)	1.77 (1.49–2.11)	Baseline	1.57 (1.30–1.90)	2.21 (1.53–3.19)	11.47 (4.50–29.21)	
^†^ Adjusted Odds ratio (95% CI)	1.73 (1.42–2.09)	Baseline	1.55 (1.25–1.90)	2.12 (1.39–3.22)	14.17 (4.45–45.09)	
Mortality						<0.001
Unadjusted Odds ratio (95% CI)	3.32 (2.55–4.33)	Baseline	2.52 (1.86–3.41)	5.49 (3.45–8.73)	28.25 (12.13–65.78)	
^†^ Adjusted Odds ratio (95% CI)	3.70 (2.73–5.00)	Baseline	2.78 (1.98–3.90)	6.79 (3.90–11.81)	52.45 (16.43–167.42)	
Readmissions						>0.05
Unadjusted Odds ratio (95% CI)	1.29 (1.05–1.59)	Baseline	1.27 (1.01–1.59)	1.26 (0.78–2.06)	3.46 (0.97–12.33)	
^†^ Adjusted Odds ratio (95% CI)	1.16 (0.93–1.44)	Baseline	1.17 (0.93–1.48)	1.19 (0.73–1.92)	0.70 (0.17–2.81)	

^†^ Model adjusted for age, sex, Charlson index, CURB-65, Hospital Frailty Risk Score (HFRS), Number of hospitalisations in previous 1 year, Number of Emergency Department presentations in last 6 months, microbiological aetiology, and antibiotics prescribed during admission. CI, confidence interval.

**Table 3 jcm-14-05273-t003:** Association of individual vital sign abnormality with composite outcome of readmission or death after multivariable regression analysis.

Vital Sign Abnormality	aOR *	95% CI	*p* Value
Temperature ≥ 37.8 °C	0.91	0.59–1.41	0.699
Cardiovascular parameters			
Heart rate ≥ 100/min	1.70	1.33–2.17	<0.001
Systolic blood pressure ≤ 90 mm Hg	4.51	2.51–8.11	<0.001
Respiratory parameters			
Respiratory rate > 24/min	3.13	2.16–4.54	<0.001
Oxygen saturation of <90%	1.54	1.08–2.20	0.017

* Model adjusted for age, sex, Charlson index, CURB-65, Hospital Frailty Risk Score (HFRS), Number of hospitalisations in previous 1 year, Number of Emergency Department presentations in last 6 months, microbiological aetiology, and antibiotics prescribed during admission. aOR adjusted odds ratio; CI confidence interval.

## Data Availability

Data available from corresponding author on reasonable request if approval is granted by the ethics committee.

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
