# Peer review of "Clinical Instability at Discharge and Post-Discharge Outcomes in Patients with Community-Acquired Pneumonia: An Observational Study"

_jcm, 2025, doi:10.3390/jcm14155273_

Round 1
Reviewer 1 Report
Comments and Suggestions for Authors
Title:
Clinical Instability at Discharge and Post-Discharge Outcomes in Patients with Community-Acquired Pneumonia: An Observational Study
Review:
The manuscript studies the clinical instability at discharge and post-discharge outcomes in patients with community-acquired pneumonia (CAP) in Australia. This issue has interest from medical assistance and some suggestions could be made in order to improve the manuscript.
- A definition of CAP could be adequate for a better understanding of the manuscript.
- A mention of the microbiological agents of CAP could be important considering that these agents could make a difference in the outcomes.
- In addition, no treatments of CAP patients have been reported in the manuscript.
- Multilevel logistic regression was employed. However, no reference level is reported for this analysis.
- Could you explain the criteria for the inclusion of potential confounding factors in the multivariable analysis?
- Mention the computer program used for the statistical analysis.
- Considering the importance of readmissions, in Table 1 it would be advisable to separate deaths and readmissions in order to better assess the results.
- Figure 2. Instability category. Could you explain this variable?
- A limitation of the study includes no information about risk factors of CAP among patients, such as socioeconomic status, smoking, specific comorbidities, and drugs that could be associated with the outcomes (1-2).
- The authors need to follow the reference guidelines of the journal.
There are some mistakes in the references. Examples:
-Reference 2: no reported volume and pages.
-Reference 3: no reported journal or editorial.
References:
1.Calvillo-King L, Arnold D, Eubank KJ, Lo M, Yunyongying P, Stieglitz H, Halm EA. Impact of social factors on risk of readmission or mortality in pneumonia and heart failure: systematic review. J Gen Intern Med. 2013; 28:269-82.
- Liapikou A, Cilloniz C, Torres A. Drugs that increase the risk of community-acquired pneumonia: a narrative review. Expert Opin Drug Saf. 2018;17:991-1003.
Author Response
Comments and Suggestions for Authors
Title:
Clinical Instability at Discharge and Post-Discharge Outcomes in Patients with Community-Acquired Pneumonia: An Observational Study
Review:
The manuscript studies the clinical instability at discharge and post-discharge outcomes in patients with community-acquired pneumonia (CAP) in Australia. This issue has interest from medical assistance and some suggestions could be made in order to improve the manuscript.
- A definition of CAP could be adequate for a better understanding of the manuscript.
Response: Thank you for this helpful suggestion. We have now included a definition of CAP in the Methods section of the manuscript to improve clarity on page 3.
“CAP was defined as the presence of symptoms such as fever, cough (with or without sputum production), or dyspnea in conjunction with radiological evidence of pulmonary infiltrate [7].”
- A mention of the microbiological agents of CAP could be important considering that these agents could make a difference in the outcomes.
Response: We have now added information about the microbilogical aetiology of CAP and have adjusted this variable in the regression analysis as per reviewer’s advice. Relevant updates have been made to the Abstract, Methods, and Results sections, as well as Tables 1–3 (please refer to pages 1, 3, and 4–8).
Abstract
“Multilevel logistic regression models were used to evaluate associations with outcomes, adjusting for age, sex, comorbidities, frailty, disease severity, microbiological aetiology, antibiotics prescribed during admission, and prior healthcare use.”
Materials and methods
“Information on the microbiological aetiology was obtained from culture results and viral polymerase chain reaction (PCR) testing.”
Results
“However, microbiological aetiology and antibiotic prescription patterns were similar between the two groups (p>0.05).”
“Multilevel regression analysis revealed that patients with any clinical instability within 24-hours prior to discharge had 1.73 times higher odds of experiencing the composite outcome (aOR 1.73, 95% CI 1.42-2.09, p<0.001), after adjusting for age, sex (reference: male), CCI, frailty (reference: non-frail), CURB65 score, previous hospitalisations in the year prior, ED visits in the 6 months preceding admission, microbiological aetiology (reference: no pathogen detected) and antibiotics prescribed (reference: penicillin), while accounting for hospital level clustering (Table 2).”
“While mortality was significantly higher among patients with clinical instability (aOR 3.70 95% CI 2.73-5.00, p<0.001) readmissions did not show a significant difference in the adjusted analysis compared to patients without clinical instability (aOR 1.16 95% CI 0.93-1.44, p>0.05), suggesting that clinical instability primarily influences mortality risk rather than readmission risk (Table 2).”
- In addition, no treatments of CAP patients have been reported in the manuscript.
Response: We have now included detailed information on the microbiological agents used in the treatment of CAP and incorporated this variable into our regression analyses. Relevant updates have been made to the Abstract, Methods, and Results sections, as well as Tables 1–3 (please refer to pages 1, 3, and 4–8).
Abstract
“Multilevel logistic regression models were used to evaluate associations with outcomes, adjusting for age, sex, comorbidities, frailty, disease severity, microbiological aetiology, antibiotics prescribed during admission, and prior healthcare use.”
Materials and methods
“Data on antibiotics prescribed for the treatment of CAP were extracted from the pharmacy records of the hospitals.”
Results
“However, microbiological aetiology and antibiotic prescription patterns were similar between the two groups (p>0.05).”
“Multilevel regression analysis revealed that patients with any clinical instability within 24-hours prior to discharge had 1.73 times higher odds of experiencing the composite outcome (aOR 1.73, 95% CI 1.42-2.09, p<0.001), after adjusting for age, sex (reference: male), CCI, frailty (reference: non-frail), CURB65 score, previous hospitalisations in the year prior, ED visits in the 6 months preceding admission, microbiological aetiology (reference: no pathogen detected) and antibiotics prescribed (reference: penicillin), accounting for hospital level clustering (Table 2).”
“While mortality was significantly higher among patients with clinical instability (aOR 3.70 95% CI 2.73-5.00, p<0.001) readmissions did not show a significant difference in the adjusted analysis compared to patients without clinical instability (aOR 1.16 95% CI 0.93-1.44, p>0.05), suggesting that clinical instability primarily influences mortality risk rather than readmission risk (Table 2).”
- Multilevel logistic regression was employed. However, no reference level is reported for this analysis.
Response: Thank you for the comment. We have now clarified the reference levels for categorical variables used in the multilevel logistic regression. Specifically, the reference categories were: male for sex, non-frail for frailty, no pathogen detected for microbilogical aetiology, and penicillin for antibiotics. These are now explicitly stated in the results section and relevant tables. The model also included a random intercept for hospital to account for hospital-level variance. Please refer to pages 7-8.
“Multilevel regression analysis revealed that patients with any clinical instability within 24-hours prior to discharge had 1.73 times higher odds of experiencing the composite outcome (aOR 1.73, 95% CI 1.42-2.09, p<0.001), after adjusting for age, sex (reference: male), CCI, frailty (reference: non-frail), CURB65 score, previous hospitalisations in the year prior, ED visits in the 6 months preceding admission, microbiological aetiology (reference: no pathogen detected) and antibiotics prescribed (reference: penicillin), accounting for hospital level clustering (Table 2).”
- Could you explain the criteria for the inclusion of potential confounding factors in the multivariable analysis?
Response: Thank you for this important question. Potential confounding factors were selected based on a combination of prior literature, clinical relevance, and biological plausibility. Variables such as age, sex, Charlson Comorbidity Index (CCI), CURB-65 score, Hospital Frailty Risk Score (HFRS), number of prior hospitalisations, ED presentations, microbiological aetilogy and antibiotic prescriptions were included due to their established associations with adverse outcomes in patients with CAP. All variables were determined a priori, and no automated variable selection methods were used, to avoid bias from data-driven selection.
We have now included this explanation with supporting references in the text. Please refer to page 4.
“Potential confounding factors were selected based on a combination of prior literature, clinical relevance, and biological plausibility. Variables such as age, sex, CCI, CURB-65 score, Hospital Frailty Risk Score (HFRS), number of prior hospitalisations, ED presentations, microbiological aetiology, and antibiotics prescriptions were included due to their established associations with adverse outcomes in patients with CAP [24-26].”
- Mention the computer program used for the statistical analysis.
Response: This is now described. Please refer to page 4.
“All statistical analysis were performed using STATA software vs 19.0 (StataCorp LLC, College Station, TX, USA) and Python.”
- Considering the importance of readmissions, in Table 1 it would be advisable to separate deaths and readmissions in order to better assess the results.
Response: Thank you for your helpful suggestion. We have now revised Table 1 to separately present 30-day mortality and 30-day readmissions, in addition to the composite outcome of death or readmission. This provides a clearer understanding of the contribution of each component to the overall outcome. The updated table includes the following new rows:
- 30-day mortality, n (%): 246 (6.1%) overall; 0 in the no-outcome group; 246 (28.1%) in the composite outcome group (p<0.001)
- 30-day readmissions, n (%): 630 (15.8%) overall; 0 in the no-outcome group; 630 (71.9%) in the composite outcome group (p<0.001)
A footnote has been added to clarify that readmissions excluded patients who died within 30 days to ensure events are counted only once within the composite measure.
- Figure 2. Instability category. Could you explain this variable?
Response: Thank you for your comment. The instability category presented in Figure 2 refers to the number of vital sign abnormalities (ranging from 0 to 3) identified within 24 hours prior to discharge. We have now clarified this in Figure 2 and its legend.
“Figure 2 Rates of 30-day adverse outcomes stratified by number of vital sign abnormalities prior to discharge.”
- A limitation of the study includes no information about risk factors of CAP among patients, such as socioeconomic status, smoking, specific comorbidities, and drugs that could be associated with the outcomes (1-2).
Response: Thank you for this insightful comment. We agree that factors such as socioeconomic status, smoking, specific comorbidities (e.g., COPD, diabetes), and certain medications have been identified in the literature as important contributors to outcomes in patients with community-acquired pneumonia (CAP). Unfortunately, due to limitations inherent in our retrospective dataset, detailed information on smoking status, individual-level socioeconomic factors, and medication use beyond antibiotics was not consistently available and could not be reliably included in our analysis. However, to partially address this, we did adjust for proxies of comorbidity burden and frailty, including the Charlson Comorbidity Index (CCI), CURB-65 score, and the Hospital Frailty Risk Score (HFRS), which have been validated as predictors of adverse outcomes in CAP populations. We have now acknowledged this additional limitation in the revised Limitations section of the manuscript as follows:
“Additionally, our dataset lacked detailed information on certain established risk factors for CAP outcomes, such as smoking status, individual socioeconomic factors, and non-antibiotic medications (e.g., proton pump inhibitors, antipsychotics) that may influence pneumonia risk or recovery [42,43].This may have limited our ability to fully adjust for all potential confounding variables.”
- The authors need to follow the reference guidelines of the journal.
There are some mistakes in the references. Examples:
-Reference 2: no reported volume and pages.
-Reference 3: no reported journal or editorial.
Response: We have now corrected references as per journal guidelines.
References:
1.Calvillo-King L, Arnold D, Eubank KJ, Lo M, Yunyongying P, Stieglitz H, Halm EA. Impact of social factors on risk of readmission or mortality in pneumonia and heart failure: systematic review. J Gen Intern Med. 2013; 28:269-82.
- Liapikou A, Cilloniz C, Torres A. Drugs that increase the risk of community-acquired pneumonia: a narrative review. Expert Opin Drug Saf. 2018;17:991-1003.
jcm-3742392-peer-review-v1
References
- Metlay, J.P.; Waterer, G.W.; Long, A.C.; Anzueto, A.; Brozek, J.; Crothers, K.; Cooley, L.A.; Dean, N.C.; Fine, M.J.; Flanders, S.A.; et al. Diagnosis and Treatment of Adults with Community-acquired Pneumonia. An Official Clinical Practice Guideline of the American Thoracic Society and Infectious Diseases Society of America. Am. J. Respir. Crit. Care Med. 2019, 200, e45-e67, doi:10.1164/rccm.201908-1581ST.
- Dagan, E.; Novack, V.; Porath, A. Adverse outcomes in patients with community acquired pneumonia discharged with clinical instability from Internal Medicine Department. Scand. J. Infect. Dis. 2006, 38, 860-866, doi:10.1080/00365540600684397.
- Cillóniz, C.; Torres, A.; Niederman, M.S. Management of pneumonia in critically ill patients. BMJ 2021, 375, e065871, doi:10.1136/bmj-2021-065871.
- Calvillo-King, L.; Arnold, D.; Eubank, K.J.; Lo, M.; Yunyongying, P.; Stieglitz, H.; Halm, E.A. Impact of social factors on risk of readmission or mortality in pneumonia and heart failure: systematic review. J. Gen. Intern. Med. 2013, 28, 269-282, doi:10.1007/s11606-012-2235-x.
- Liapikou, A.; Cilloniz, C.; Torres, A. Drugs that increase the risk of community-acquired pneumonia: a narrative review. Expert Opin Drug Saf 2018, 17, 991-1003, doi:10.1080/14740338.2018.1519545.
Reviewer 2 Report
Comments and Suggestions for Authors
Thank you for the opportunity to review this well-constructed and clinically meaningful manuscript.
This study evaluated the association between clinical instability within 24 hours prior to discharge and 30-day outcomes (readmission or death) among hospitalized patients with community-acquired pneumonia (CAP). Using a retrospective cohort design and adjusting for key confounders such as frailty and disease severity (HFRS and CURB-65), the study provides valuable insights into a clinically relevant topic. The inclusion of frailty and disease severity, which were not adequately considered in prior research, is a particular strength.
However, several points require clarification and improvement to enhance the transparency and reproducibility of the study.
Overall Assessment
This study provides meaningful contributions to the literature on CAP discharge planning and post-discharge outcomes. The inclusion of frailty and disease severity adjustments is commendable. However, to strengthen the manuscript's methodological transparency and reproducibility, clarification of the multilevel modeling strategy, missing data handling, and reconsideration of p-values in descriptive tables are needed. Addressing these issues will further enhance the clarity and impact of this valuable study.
Comments
1. Incomplete description of the multilevel logistic regression model
(Reference: p.4, lines 116–120)
"Multilevel logistic regression was employed ... along with hospital-level variance."
While the authors mention the use of multilevel logistic regression, it is not stated whether a random intercept or random slope model was used. In addition, although “hospital-level variance” is noted, no statistical estimates (e.g., variance components, ICC, or p-values) are presented in the results section. Therefore, the justification for using a hierarchical model remains unclear, and the model specification should be described more explicitly.
2. Missing data handling not reported
(Reference: p.3–5)
Although exclusion criteria are provided (e.g., death, transfer to other hospitals), there is no information regarding the amount or handling of missing data, particularly for vital signs or covariates. The authors mention that cases with missing vital signs were excluded (lines 104-105), but the number of excluded cases and their characteristics are not reported. This should be addressed to ensure full transparency in sample construction.
3. Use of p-values in descriptive Table 1
(Reference: p.5–6, Table 1)
Table 1 presents numerous p-values comparing baseline characteristics by outcome status (death or readmission). However, according to the STROBE statement, p-values for baseline comparisons are not recommended in observational studies, that is those without random assignment. Descriptive statistics should be presented without p-values to avoid misinterpretation of statistical significance.
4. Potential under-ascertainment of readmissions
(Reference: p.4, line 110)
“Readmissions were tracked for any public hospital within South Australia.”
Readmissions were limited to public hospitals within a single region. This may lead to under-ascertainment if patients were readmitted to private or out-of-state facilities. This limitation should be explicitly acknowledged and discussed.
5. Clarification on consistency of hierarchical modeling across analyses (optional but encouraged)
While the main logistic regression analyses accounted for hospital-level variance through a multilevel structure, it is unclear whether the Fine and Gray competing risk model incorporated a similar hierarchical framework. I understand that including random effects in competing risk models can be technically challenging and not always feasible, especially with limited clusters or small event counts. However, if feasible, aligning the modeling structure across analyses—or at least briefly discussing this limitation and its implications—would improve methodological coherence.
Author Response
Comments and Suggestions for Authors
Thank you for the opportunity to review this well-constructed and clinically meaningful manuscript.
This study evaluated the association between clinical instability within 24 hours prior to discharge and 30-day outcomes (readmission or death) among hospitalized patients with community-acquired pneumonia (CAP). Using a retrospective cohort design and adjusting for key confounders such as frailty and disease severity (HFRS and CURB-65), the study provides valuable insights into a clinically relevant topic. The inclusion of frailty and disease severity, which were not adequately considered in prior research, is a particular strength.
However, several points require clarification and improvement to enhance the transparency and reproducibility of the study.
Overall Assessment
This study provides meaningful contributions to the literature on CAP discharge planning and post-discharge outcomes. The inclusion of frailty and disease severity adjustments is commendable. However, to strengthen the manuscript's methodological transparency and reproducibility, clarification of the multilevel modeling strategy, missing data handling, and reconsideration of p-values in descriptive tables are needed. Addressing these issues will further enhance the clarity and impact of this valuable study.
Comments
- Incomplete description of the multilevel logistic regression model
(Reference: p.4, lines 116–120)
"Multilevel logistic regression was employed ... along with hospital-level variance."
While the authors mention the use of multilevel logistic regression, it is not stated whether a random intercept or random slope model was used. In addition, although “hospital-level variance” is noted, no statistical estimates (e.g., variance components, ICC, or p-values) are presented in the results section. Therefore, the justification for using a hierarchical model remains unclear, and the model specification should be described more explicitly.
Response:
Thank you for this helpful comment. We have clarified in the Methods section (page 4) that a random intercept multilevel logistic regression model was used, with hospital specified as the clustering variable to account for potential differences in care practices and outcomes across sites.
As requested, we have also reported the variance component, intraclass correlation coefficient (ICC), and the likelihood ratio (LR) test comparing the multilevel model to a standard logistic regression model in the Results section (page 9). The estimated hospital-level variance was negligible (3.35 × 10⁻³⁵), and the ICC was near zero, indicating minimal clustering of outcomes by hospital. The LR test (χ² = 0.00) showed no improvement in model fit, suggesting that the multilevel structure did not contribute additional explanatory power.
We have now included this in the methods and results section on pages 4 and 9.
Materials and methods
“A random intercept multilevel logistic regression model was employed to assess the association between clinical instability and the composite outcome, with hospital specified as a clustering variable to account for potential differences in patient populations, care practices, and outcome rates across sites. The model was adjusted for potential confounders including age, sex, Charlson comorbidity index (CCI), CURB-65 score, HFRS, number of hospital admissions in the previous 1 year, number of ED presentations in last 6 months, microbiological aetiology, and antibiotics prescriptions. These covariates were selected based on prior literature, clinical relevance, and biological plausibility, given their established associations with adverse outcomes in patients with CAP [24-26].
To evaluate the appropriateness of the multilevel structure, we calculated the hospital-level variance component and the intraclass correlation coefficient (ICC). These measures quantified the proportion of total outcome variance attributable to between-hospital differences.”
Results
“Multilevel regression analysis revealed that patients with any clinical instability within 24-hours prior to discharge had 1.73 times higher odds of experiencing the composite outcome (aOR 1.73, 95% CI 1.42-2.09, p<0.001), after adjusting for age, sex (reference: male), CCI, frailty (reference: non-frail), CURB65 score, previous hospitalisations in the year prior, ED visits in the 6 months preceding admission, microbiological aetiology (reference: no pathogen detected) and antibiotics prescribed (reference: penicillin), while accounting for hospital level clustering (Table 2). The estimated hospital-level variance was negligible (variance = 3.35e–35), with an ICC near zero. The likelihood ratio test comparing the multilevel model to a standard logistic regression showed no improvement in model fit (χ² = 0.00), indicating that outcomes were not meaningfully clustered by hospital.”
- Missing data handling not reported
(Reference: p.3–5)
Although exclusion criteria are provided (e.g., death, transfer to other hospitals), there is no information regarding the amount or handling of missing data, particularly for vital signs or covariates. The authors mention that cases with missing vital signs were excluded (lines 104-105), but the number of excluded cases and their characteristics are not reported. This should be addressed to ensure full transparency in sample construction.
Response:
Thank you for this important point. We have clarified the flow of patient inclusion and handling of missing data in both the Methods and Results sections. Of the 6,319 patients initially admitted with CAP, 1,642 were excluded due to in-hospital death, transfers, and self-discharges. Among the remaining 4,677 patients, 693 (14.8%) were excluded due to missing vital sign data within 24 hours prior to discharge. This resulted in a final analytic sample of 3,984 patients. Among these, 374 (9.4%) had missing data for antibiotic prescriptions. All other regression covariates were complete. A complete-case analysis was conducted, and no imputation was performed. We have also updated Figure 1 to reflect this in the flow diagram.
Results
“Over a 4-year period, 6,319 adults with CAP were admitted to the two hospitals (Figure 1). After applying exclusion criteria (including in-hospital deaths, transfer to another hospital, hospital at home, and self-discharge), 4,677 patients remained. Of these, 693 patients (14.8%) had missing data on vital signs in the 24 hours prior to discharge and were excluded. The final analytic sample comprised 3,984 patients, of whom 813 (20.4%) experienced clinical instability within 24 hours of discharge (Figure 1). Among those included, 374 patients (9.4%) had missing data for antibiotic prescriptions, and all other covariates were complete.”
- Use of p-values in descriptive Table 1
(Reference: p.5–6, Table 1)
Table 1 presents numerous p-values comparing baseline characteristics by outcome status (death or readmission). However, according to the STROBE statement, p-values for baseline comparisons are not recommended in observational studies, that is those without random assignment. Descriptive statistics should be presented without p-values to avoid misinterpretation of statistical significance.
Response: Thank you for highlighting this. In accordance with the STROBE statement and best practices for reporting observational studies, we have removed p-values from Table 1. The table now presents descriptive statistics (means with standard deviations, medians with interquartile ranges, and proportions) stratified by outcome status to allow the reader to assess the clinical relevance of group differences without overemphasis on statistical significance. This change has been made in Table 1 (pages 7-8).
- Potential under-ascertainment of readmissions
(Reference: p.4, line 110)
“Readmissions were tracked for any public hospital within South Australia.”
Readmissions were limited to public hospitals within a single region. This may lead to under-ascertainment if patients were readmitted to private or out-of-state facilities. This limitation should be explicitly acknowledged and discussed.
Response:Thank you for this important observation. We acknowledge that tracking readmissions was limited to public hospitals within South Australia, and therefore may have missed readmissions to private hospitals or facilities outside the state. While the majority of patients in our cohort are typically managed within the public health system, we agree that this may result in some under-ascertainment of readmission events. We have now included this as a limitation in the Discussion section on page 13, noting the potential for incomplete capture of all readmissions.
“Readmissions were only captured within the South Australian public hospital system. This may have led to under-ascertainment of readmission events among patients who were admitted to private hospitals or facilities outside the state, potentially underestimating the true rate of readmissions.”
- Clarification on consistency of hierarchical modeling across analyses (optional but encouraged)
While the main logistic regression analyses accounted for hospital-level variance through a multilevel structure, it is unclear whether the Fine and Gray competing risk model incorporated a similar hierarchical framework. I understand that including random effects in competing risk models can be technically challenging and not always feasible, especially with limited clusters or small event counts. However, if feasible, aligning the modeling structure across analyses—or at least briefly discussing this limitation and its implications—would improve methodological coherence.
Response: Thank you for this thoughtful suggestion. We confirm that the Fine and Gray competing risk analysis did not incorporate a multilevel or random effects structure. This was due to the technical limitations of implementing hierarchical models within competing risk frameworks, particularly with only two hospital clusters. We agree that this represents a methodological inconsistency.
References
- Dagan, E.; Novack, V.; Porath, A. Adverse outcomes in patients with community acquired pneumonia discharged with clinical instability from Internal Medicine Department. Scand. J. Infect. Dis. 2006, 38, 860-866, doi:10.1080/00365540600684397.
- Sharma, Y.; Mangoni, A.A.; Shahi, R.; Horwood, C.; Thompson, C. Recent temporal trends, characteristics and outcomes of patients with non-COVID-19 community-acquired pneumonia at two tertiary hospitals in Australia: an observational study. Intern. Med. J. 2024, 54, 1686-1693, doi:10.1111/imj.16469.
- Cillóniz, C.; Torres, A.; Niederman, M.S. Management of pneumonia in critically ill patients. BMJ 2021, 375, e065871, doi:10.1136/bmj-2021-06587
Reviewer 3 Report
Comments and Suggestions for Authors
jcm-3742392-peer-review-v1
Clinical Instability at Discharge and Post-Discharge Outcomes 2 in Patients with Community-Acquired Pneumonia: An Observa-3 tional Study
This article investigates whether clinical instability —defined by abnormal vital signs —within 24 hours before hospital discharge predicts 30-day mortality and readmission in patients with community-acquired pneumonia The study is significant as it highlights the importance of discharge readiness and suggests that unresolved clinical instability substantially increases short-term mortality risk, underscoring the need for better discharge protocols in community-acquired pneumonia care.
INTRODUCTION
It is recommended that you incorporate the most recent discharge guidelines into your introduction and later in the discussion.
- Dutch Working Party on Antibiotic Policy (SWAB)/NVALT CAP Guidelines 2024 https://adult.nl.antibiotica.app/sites/default/files/2024-07/SWAB%20guideline%20CAP%202024%20020724_def.pdf
- NICE Guideline on Pneumonia (Draft for consultation, April 2025) https://www.nice.org.uk/guidance/indevelopment/gid-ng10357
- MEDSCAPE Community-Acquired Pneumonia (CAP) https://emedicine.medscape.com/article/234240-overview
- Intermountain health https://intermountainhealthcare.org/ckr-ext/Dcmnt?ncid=520102603
MATERIAL AND METHODS
Line 80: “between between” duplicated word. Should be “between”.
Clarify if the CURB-65 score was calculated prospectively or retrospectively using EMR data (line 86).
Indicate how missing data were handled—especially in line 104 regarding exclusion due to missing vitals.
The definition of frailty (HFRS ≥5) is mentioned, but rationale or reference supporting this cutoff should be emphasized more clearly.
Specify the ICD-10-AM codes used for CAP in Supplementary Material or Appendix for transparency.
RESULTS
Add p-value for comparisons in Figures 2 or refer explicitly to them in text.
Clarify sample derivation in Figure 1—e.g., exact reasons for exclusion of ~2,335 patients.
Interpretation of adjusted vs. unadjusted ORs in Table 2 should be mentioned briefly in text for clarity.
In Table 3, organize the vital signs by physiological system—for example, grouping cardiovascular parameters (e.g., blood pressure, heart rate) separately from respiratory parameters (e.g., respiratory rate, oxygen saturation). This categorization could enhance readability and help the reader more easily interpret which types of clinical instability are most strongly associated with adverse outcomes.
DISCUSSION
Include A more structured discussion of strengths (e.g. , large sample size, multicenter design).
Lines 227–229 mention factors like discharge planning—integrate this with existing literature on post-discharge interventions.
Avoid speculative explanations without citations, such as the adaptation of healthcare teams during the pandemic (lines 244–248), unless supported by references.
Line 286–290: The issue of patients discharged on palliative care should be quantified if possible, or clarified as a limitation in more detail.
BIBLIOGRAPHY
Line 331: Missing volume/issue/page numbers in reference 2.
Reference 3 lacks journal name.
Author Response
Clinical Instability at Discharge and Post-Discharge Outcomes 2 in Patients with Community-Acquired Pneumonia: An Observa-3 tional Study
This article investigates whether clinical instability —defined by abnormal vital signs —within 24 hours before hospital discharge predicts 30-day mortality and readmission in patients with community-acquired pneumonia The study is significant as it highlights the importance of discharge readiness and suggests that unresolved clinical instability substantially increases short-term mortality risk, underscoring the need for better discharge protocols in community-acquired pneumonia care.
INTRODUCTION
It is recommended that you incorporate the most recent discharge guidelines into your introduction and later in the discussion.
- Dutch Working Party on Antibiotic Policy (SWAB)/NVALT CAP Guidelines 2024 https://adult.nl.antibiotica.app/sites/default/files/2024-07/SWAB%20guideline%20CAP%202024%20020724_def.pdf
- NICE Guideline on Pneumonia (Draft for consultation, April 2025) https://www.nice.org.uk/guidance/indevelopment/gid-ng10357
- MEDSCAPE Community-Acquired Pneumonia (CAP) https://emedicine.medscape.com/article/234240-overview
- Intermountain health https://intermountainhealthcare.org/ckr-ext/Dcmnt?ncid=520102603
Response: We thank the reviewer for this suggestion and have now included these recent guidelines in the Introduction and Discussion sections on pages 2 and 11.
“For instance, the 2024 Dutch Working Party on Antibiotic Policy (SWAB)/NVALT CAP guideline [9], the NICE guideline on pneumonia [10], and clinical frameworks such as those from Intermountain Health [11] and Medscape [12] all emphasise ensuring physiological stability and appropriate clinical recovery before discharge. These guidelines increasingly recognise the importance of not only antimicrobial optimisation but also holistic assessment of discharge fitness based on vital signs, functional status, and comorbidities.”
“Our findings support current clinical guidelines recommending that discharge decisions be informed by vital sign stability. Recent international guidelines highlight objective resolution of vital sign abnormalities—especially respiratory rate and blood pressure—as essential for safe discharge decisions [9-11].However, as readmissions may be less influenced by physiologic measures, multifactorial post-discharge strategies—including enhanced discharge planning, patient education, early follow-up, and community supports—are likely more effective at preventing readmissions [32,41]. These results reinforce the growing consensus that bundled transitional care interventions can reduce readmission risk more effectively than discharge criteria alone.”
MATERIAL AND METHODS
Line 80: “between between” duplicated word. Should be “between”.
Response: We have now corrected this error.
Clarify if the CURB-65 score was calculated prospectively or retrospectively using EMR data (line 86).
Response: Thank you for this comment. We have now clarified in the Methods section that the CURB-65 score was calculated retrospectively using data extracted from the electronic medical records. Please refer to page 3.
“Pneumonia severity was assessed using the CURB-65 score which was retrospectively calculated based on clinical and laboratory data obtained from the EMR [20].”
Indicate how missing data were handled—especially in line 104 regarding exclusion due to missing vitals.
Response: Thank you for pointing this out. We have now clarified in the manuscript how missing data were handled. Specifically, patients with missing vital sign data within 24 hours prior to discharge were excluded from the analysis, as indicated in the study flowchart. We have now documented this on page 4.
“Patients who lacked documentation of any vital signs within 24 hours prior to discharge were excluded from the analysis. This exclusion is detailed in the study flowchart.”
The definition of frailty (HFRS ≥5) is mentioned, but rationale or reference supporting this cutoff should be emphasized more clearly.
Response: We have now provided a reference supporting this cutoff.
Specify the ICD-10-AM codes used for CAP in Supplementary Material or Appendix for transparency.
Response: Thank you for the suggestion. We agree that providing the specific ICD-10-AM codes enhances the transparency and reproducibility of our study. The ICD-10-AM codes used to identify CAP cases have now been listed in the Supplementary TableS2.
RESULTS
Add p-value for comparisons in Figures 2 or refer explicitly to them in text.
Response: We have now explicitly explained the comparisons with p values in reference to Figure 2 in the text. Please refer to page
“The likelihood of the composite outcome increased with the number of vital sign abnormalities; 73.9% of patients with three or more vital sign abnormalities experienced a composite outcome, compared to 28.0% of those with only one abnormality (p<0.001) (Figure 2). While 30-day mortality also rose significantly with increasing number of vital sign abnormalities (p<0.001), no significant difference was observed in readmission rates across groups (p>0.05) (Figure 2).”
Clarify sample derivation in Figure 1—e.g., exact reasons for exclusion of ~2,335 patients.
Response: We appreciate the reviewer’s suggestion. We would like to clarify that the reasons for exclusion of the 2,335 patients—namely, self-discharged (n=128), transferred to other hospitals (n=697), discharged to hospital-at-home services (n=325), in-hospital deaths (n=492), and missing data on vital signs (n=693)—are already detailed in Figure 1.
Interpretation of adjusted vs. unadjusted ORs in Table 2 should be mentioned briefly in text for clarity.
Response: We thank the reviewer for this comment and have clarified adjusted and unadjusted ORs in the text. Please refer to page 8.
“Multilevel regression analysis revealed that patients with any clinical instability within 24-hours prior to discharge had 1.73 times higher odds of experiencing the composite outcome (aOR 1.73, 95% CI 1.42-2.09, p<0.001), after adjusting for age, sex (reference: male), CCI, frailty (reference: non-frail), CURB65 score, previous hospitalisations in the year prior, ED visits in the 6 months preceding admission, microbiological aetiology (reference: no pathogen detected) and antibiotics prescribed (reference: penicillin), while accounting for hospital level clustering (Table 2).”
“While mortality was significantly higher among patients with clinical instability (aOR 3.70 95% CI 2.73-5.00, p<0.001) readmissions did not show a significant difference in the adjusted analysis compared to patients without clinical instability (aOR 1.16 95% CI 0.93-1.44, p>0.05), suggesting that clinical instability primarily influences mortality risk rather than readmission risk (Table 2).”
In Table 3, organize the vital signs by physiological system—for example, grouping cardiovascular parameters (e.g., blood pressure, heart rate) separately from respiratory parameters (e.g., respiratory rate, oxygen saturation). This categorization could enhance readability and help the reader more easily interpret which types of clinical instability are most strongly associated with adverse outcomes.
Response: We have now organised the vital signs by physiological system, grouping cardiovascular and respiratory parameters separately in Table 3, as recommended by the reviewer.
DISCUSSION
Include A more structured discussion of strengths (e.g. , large sample size, multicenter design).
Response: We have now structured the Discussion section by adding headings and a summary of the study’s strengths, as recommended by the reviewer. Please refer to pages 10 and 12.
“This multicentre cohort study of nearly 4000 patients hospitalised with CAP provides strong evidence that clinical instability in the 24 hours prior to discharge is associated with significantly higher odds of composite outcome of 30-day death or readmission. Importantly, the risk of adverse outcomes increased progressively with the number of abnormal vital signs at discharge, reinforcing the value of clinical stability assessments in discharge planning.”
“Strengths
Key strengths of this study include its large sample size, inclusion of multiple hospitals, and the use of comprehensive linked data that captured both in-hospital and post-discharge events across institutions. Unlike prior studies, we adjusted for frailty and disease severity and included a wide range of clinical variables, improving the validity of our findings.”
Lines 227–229 mention factors like discharge planning—integrate this with existing literature on post-discharge interventions.
Response: We have now included a separate paragraph on discharge planning and have discussed post-discharge interventions as per reviewer’s suggestions. Please refer to page 11.
Implications for Discharge Planning and Post-Discharge Care
“Our findings support current clinical guidelines recommending that discharge decisions be informed by vital sign stability. Recent international guidelines highlight objective resolution of vital sign abnormalities—especially respiratory rate and blood pressure—as essential for safe discharge decisions [9-11].However, as readmissions may be less influenced by physiologic measures, multifactorial post-discharge strategies—including enhanced discharge planning, patient education, early follow-up, and community supports—are likely more effective at preventing readmissions [33,42]. These results reinforce the growing consensus that bundled transitional care interventions can reduce readmission risk more effectively than discharge criteria alone.”
Avoid speculative explanations without citations, such as the adaptation of healthcare teams during the pandemic (lines 244–248), unless supported by references.
Response: We thank the reviewer for the comment and have now added two supporting references. Please refer to page 11.
“They suggest that, despite the overwhelming pressure on hospital systems and the drive for efficiency during the pandemic, the reduction in LOS did not result in compromised discharge practices. It is likely that healthcare teams adapted their practices effectively to ensure safe and appropriate discharge planning, even under the strain of increased hospital admissions and constrained resources [40,41].”
Line 286–290: The issue of patients discharged on palliative care should be quantified if possible, or clarified as a limitation in more detail.
Response: The issue of palliative patients has now been clarified in the limitations section on page 12.
“Third, we were unable to identify patients who were discharged with palliative intent, particularly those transferred to nursing homes, who likely had a higher baseline risk of death.”
BIBLIOGRAPHY
Line 331: Missing volume/issue/page numbers in reference 2.
Response: The reference 2 has now been corrected.
Reference 3 lacks journal name.
Response: This error has been rectified.
References
- The Dutch Working Party on Antibiotic Policy (SWAB) and Dutch Association of Chest Physicians (NVALT). Management of Community-Acquired Pneumonia in Adults: The 2024 Practice Guideline. SWAB/NVALT; 2024.
- National Institute for Health and Care Excellence (NICE). Pneumonia in adults: diagnosis and management. NICE Guideline NG138; NICE: London, UK, 2023.
- Intermountain Health. Diagnosis and Management of Community-Acquired Pneumonia in Adults. Intermountain Clinical Guidelines, 2025.
- Medscape. Community-Acquired Pneumonia (CAP); 2025. Available online: https://emedicine.medscape.com/article/234240-overview.
- Jasti, H.; Mortensen, E.M.; Obrosky, D.S.; Kapoor, W.N.; Fine, M.J. Causes and risk factors for rehospitalization of patients hospitalized with community-acquired pneumonia. Clin. Infect. Dis. 2008, 46, 550-556, doi:10.1086/526526.
- Yoon, S.; Mo, J.; Lim, Z.Y.; Lu, S.Y.; Low, S.G.; Xu, B.; Loo, Y.X.; Koh, C.W.; Kong, L.Y.; Towle, R.M.; et al. Impact of COVID-19 Measures on Discharge Planning and Continuity of Integrated Care in the Community for Older Patients in Singapore. Int J Integr Care 2022, 22, 13, doi:10.5334/ijic.6416.
- Ilg, A.; Moskowitz, A.; Konanki, V.; Patel, P.V.; Chase, M.; Grossestreuer, A.V.; Donnino,
M.W. Performance of the CURB-65 Score in Predicting Critical Care Interventions in Patients
Admitted With Community-Acquired Pneumonia. Ann. Emerg. Med. 2019, 74, 60-68,
doi:10.1016/j.annemergmed.2018.06.017.
- Naylor, M.D.; Brooten, D.; Campbell, R.; Jacobsen, B.S.; Mezey, M.D.; Pauly, M.V.; Schwartz, J.S. Comprehensive discharge planning and home follow-up of hospitalized elders: a randomized clinical trial. JAMA 1999, 281, 613-620, doi:10.1001/jama.281.7.613.
- Calvillo-King, L.; Arnold, D.; Eubank, K.J.; Lo, M.; Yunyongying, P.; Stieglitz, H.; Halm, E.A. Impact of social factors on risk of readmission or mortality in pneumonia and heart failure: systematic review. J. Gen. Intern. Med. 2013, 28, 269-282, doi:10.1007/s11606-012-2235-x.
- Haldane, V.; De Foo, C.; Abdalla, S.M.; Jung, A.S.; Tan, M.; Wu, S.; Chua, A.; Verma, M.; Shrestha, P.; Singh, S.; et al. Health systems resilience in managing the COVID-19 pandemic: lessons from 28 countries. Nat. Med. 2021, 27, 964-980, doi:10.1038/s41591-021-01381-y.
Reviewer 4 Report
Comments and Suggestions for Authors
I would like to congratulate the authors for their interesting and relevant manuscript.
- Clarity, Relevance, and Structure
The manuscript is clear, relevant, and well-structured. The language is generally clear, and the flow is logical. Definitions (e.g., clinical instability, CURB-65, HFRS) are introduced early, aiding understanding. The topic is highly relevant in the context of hospital discharge practices, post-discharge outcomes, and pneumonia care, especially post-COVID. The manuscript follows a standard scientific format: Abstract, Introduction, Methods, Results, Discussion, and Conclusion. Each section is appropriately detailed.
Minor suggestion: Some redundancies and typographic inconsistencies (e.g., line 80: "between between") should be corrected.
- References: the manuscript cites recent and relevant studies, mostly from 2019–2024. Self-citation is present but not excessive, given that the authors have published in this field, referencing their related studies (e.g., on CAP or frailty) is reasonable and justified.
- Scientific soundness and experimental design: the manuscript is scientifically sound and employs an appropriate retrospective cohort design for the research question. It is a retrospective analysis using EMR data from two tertiary hospitals over four years. Uses large cohort sample (n=3,984 after exclusions), with defined inclusion/exclusion criteria.
Limitation (acknowledged): the retrospective nature and potential for unmeasured confounders.
- The methods section provides sufficient detail to reproduce the study.
- Figures, Tables, and Data Interpretation
Figures/Tables:
- Figure 1 (Study flowchart), Figure 2 (outcomes by instability count), and Figure 3 (CIF plot) are appropriate and visually clear.
- Tables 1–3 present statistical findings in an interpretable format with proper labeling.
Interpretation:
- Data interpretation is appropriate and consistent, especially regarding the distinction between mortality and readmission risks.
- Adjusted ORs and CIs are reported transparently.
- The authors avoid overclaiming, noting where findings are non-significant.
- Consistency of conclusions with results: the conclusions are fully consistent with the data presented. The authors correctly emphasize that mortality, not readmission, is the primary driver of adverse outcomes related to discharge instability. They appropriately call for attention to discharge protocols and consideration of frailty in planning transitions of care.
- Ethics and data availability statements:
- Ethical approval was obtained from the Central Adelaide Local Health Network Clinical Ethics Committee.
- Waiver of patient consent is justified for a retrospective study.
Author Response
Reviewer 4
I would like to congratulate the authors for their interesting and relevant manuscript.
- Clarity, Relevance, and Structure
The manuscript is clear, relevant, and well-structured. The language is generally clear, and the flow is logical. Definitions (e.g., clinical instability, CURB-65, HFRS) are introduced early, aiding understanding. The topic is highly relevant in the context of hospital discharge practices, post-discharge outcomes, and pneumonia care, especially post-COVID. The manuscript follows a standard scientific format: Abstract, Introduction, Methods, Results, Discussion, and Conclusion. Each section is appropriately detailed.
Response: We thank the reviewer for their comments.
Minor suggestion: Some redundancies and typographic inconsistencies (e.g., line 80: "between between") should be corrected.
Response: This error has now been corrected.
- References: the manuscript cites recent and relevant studies, mostly from 2019–2024. Self-citation is present but not excessive, given that the authors have published in this field, referencing their related studies (e.g., on CAP or frailty) is reasonable and justified.
- Scientific soundness and experimental design: the manuscript is scientifically sound and employs an appropriate retrospective cohort design for the research question. It is a retrospective analysis using EMR data from two tertiary hospitals over four years. Uses large cohort sample (n=3,984 after exclusions), with defined inclusion/exclusion criteria.
Limitation (acknowledged): the retrospective nature and potential for unmeasured confounders.
- The methods section provides sufficient detail to reproduce the study.
- Figures, Tables, and Data Interpretation
Figures/Tables:
- Figure 1 (Study flowchart), Figure 2 (outcomes by instability count), and Figure 3 (CIF plot) are appropriate and visually clear.
- Tables 1–3 present statistical findings in an interpretable format with proper labeling.
Interpretation:
- Data interpretation is appropriate and consistent, especially regarding the distinction between mortality and readmission risks.
- Adjusted ORs and CIs are reported transparently.
- The authors avoid overclaiming, noting where findings are non-significant.
- Consistency of conclusions with results: the conclusions are fully consistent with the data presented. The authors correctly emphasize that mortality, not readmission, is the primary driver of adverse outcomes related to discharge instability. They appropriately call for attention to discharge protocols and consideration of frailty in planning transitions of care.
- Ethics and data availability statements:
- Ethical approval was obtained from the Central Adelaide Local Health Network Clinical Ethics Committee.
- Waiver of patient consent is justified for a retrospective study.
Round 2
Reviewer 1 Report
Comments and Suggestions for Authors
The authors have addressed all the suggestions in our review.
However, in response to the review, the authors mention that "We have now added information about the microbilogical aetiology of CAP and have adjusted this variable in the regression analysis as per reviewer’s advice."
In addition, the authors mention several pages and Tables where information about microbiological information is presented: Abstract "microbiological aetiology," Material and Methods "Information on the microbiological aetiology was obtained from culture results and viral polymerase chain reaction (PCR) testing.” Results "However, microbiological aetiology..", and "microbiological aetiology (reference: no pathogen detected)", Table 2 "microbiological aetiology (reference: no pathogen detected)"
All this information are not included in the manuscript that the authors reviewed.
Could you explain this situation?
Author Response
Comments and Suggestions for Authors
The authors have addressed all the suggestions in our review.
However, in response to the review, the authors mention that "We have now added information about the microbilogical aetiology of CAP and have adjusted this variable in the regression analysis as per reviewer’s advice."
In addition, the authors mention several pages and Tables where information about microbiological information is presented: Abstract "microbiological aetiology," Material and Methods "Information on the microbiological aetiology was obtained from culture results and viral polymerase chain reaction (PCR) testing.” Results "However, microbiological aetiology..", and "microbiological aetiology (reference: no pathogen detected)", Table 2 "microbiological aetiology (reference: no pathogen detected)"
All this information are not included in the manuscript that the authors reviewed.
Could you explain this situation?
Response: We thank the reviewer for their careful assessment and helpful comments. We sincerely apologise for the oversight — it appears that the incorrect version of the revised manuscript was inadvertently submitted. The version reviewed did not include the updates regarding microbiological aetiology as described in our response. We have now submitted the correct, updated manuscript that includes the additional information on microbiological aetiology of CAP, including relevant references in the Abstract, Methods, Results, and Tables 1-3. We appreciate the opportunity to rectify this error and thank the reviewer again for their diligence.